# A Review of Plant Extracts and Plant-Derived Natural Compounds in the Prevention/Treatment of Neonatal Hypoxic-Ischemic Brain Injury

**DOI:** 10.3390/ijms22020833

**Published:** 2021-01-15

**Authors:** Hadi Mohsenpour, Mirko Pesce, Antonia Patruno, Azam Bahrami, Pardis Mohammadi Pour, Mohammad Hosein Farzaei

**Affiliations:** 1Department of Pediatrics, Imam Reza Hospital, Kermanshah University of Medical Sciences, Kermanshah 75333–67427, Iran; h.mohsenpour@kums.ac.ir; 2Department of Medicine and Aging Sciences, University G. d’Annunzio, 66100 Chieti, Italy; 3Medical Technology Research Center, Health Technology Institute, Kermanshah University of Medical Sciences, Kermanshah 67158-47141, Iran; azambahrami@yahoo.com; 4Department of Pharmacognosy, School of Pharmacy and Pharmaceutical Sciences, Isfahan University of Medical Sciences, Isfahan 81746-73461, Iran; p.mohamadi90@yahoo.com

**Keywords:** hypoxic-ischemic encephalopathy, neonatal hypoxic-ischemic brain damage, birth asphyxia, perinatal hypoxia-ischemia, neonates, newborns, neuroprotective, plant extracts, natural compounds

## Abstract

Neonatal hypoxic-ischemic (HI) brain injury is one of the major drawbacks of mortality and causes significant short/long-term neurological dysfunction in newborn infants worldwide. To date, due to multifunctional complex mechanisms of brain injury, there is no well-established effective strategy to completely provide neuroprotection. Although therapeutic hypothermia is the proven treatment for hypoxic-ischemic encephalopathy (HIE), it does not completely chang outcomes in severe forms of HIE. Therefore, there is a critical need for reviewing the effective therapeutic strategies to explore the protective agents and methods. In recent years, it is widely believed that there are neuroprotective possibilities of natural compounds extracted from plants against HIE. These natural agents with the anti-inflammatory, anti-oxidative, anti-apoptotic, and neurofunctional regulatory properties exhibit preventive or therapeutic effects against experimental neonatal HI brain damage. In this study, it was aimed to review the literature in scientific databases that investigate the neuroprotective effects of plant extracts/plant-derived compounds in experimental animal models of neonatal HI brain damage and their possible underlying molecular mechanisms of action.

## 1. Introduction

Hypoxia-ischemia (HI) is regarded as one of the major drawbacks of brain damage in the neonates and infants. Neonatal hypoxic-ischemic brain damage (HIBD) (synonymous with hypoxic-ischemic encephalopathy (HIE)) is the main drawback of newborn deaths and irretrievable and long-lasting neurodevelopmental disabilities in the sufferers [1]. HIE complications estimate 23% of infant deaths all over the world, and have impacted 0.7–1.2 million newborns per annum [2]. Cerebral palsy, epilepsy, mental retardation, motor and cognitive deficits, learning and behavioral disabilities, and other severe neurological disorders were regularly observed in the sufferers based on brain injury grade [3,4]. In infants with mild HIE, the death probability is 10% and the risk of neurodevelopmental disorders is 30%, while 60% of infants with severe HIE are at risk of death and all sufferers risk exposure to remarkable disabilities.

The neonatal HI (NHI) incidence is near 1–8 of every one thousand live full-term births in developed countries, and nearly 25 in one thousand live full-term births in developing countries [5,6]. Lately, advanced neonatology has evidenced a great improvement in survival of seriously premature newborns. The rate of incidence in the very low birth weight infants is 60% of all live births.

NHI commonly occurs as a result of prenatal or birth asphyxia near the time of birth [7,8]. The ordinary birth process that is correlated with asphyxia and severe asphyxia may affect a pre-existing normal brain, prompting acute encephalopathy. Severe asphyxia can arise for numerous causes, including abnormal uterine contractions, abruption of the placenta, umbilical cord compression, or failure of the neonate to initiate breathing successfully. Infection, growth retardation, and fever can enhance the brain sensitivity to nonpathogenic asphyxia. On the other hand, the prematurity subject is even more complicated which is frequently concomitant with impaired fetal growth and infection. Also, the general handling of the severe premature infants causes bases of postnatal insults. The most common form of perinatal injury is systemic asphyxia, which impacts the neonate/fetus cardiovascular system and results in a hypoperfusion by hypoxemic blood. Consequently, such injury is discussed to be reperfusion/hypoperfusion damage [9].

On the cellular status, HI actuates the activation of complex biochemical reactions, consisting of switching from oxidative to anaerobic metabolism, cellular energy failure, excitotoxicity, dysfunction of mitochondria, overload of intracellular calcium, production of free radicals, and inflammatory responses, which results in significant neuronal impairment and ultimately contributes to change in degrees of brain infarction, BBB (blood–brain barrier) dysfunction, and brain edema, finally resulting in irreversible brain injury.

The neurologic disorders extremely dwindle the quality of life in HIBD kids and escalate the socioeconomic burden on caregivers, families, and society. Currently, hypothermia is the only standard treatment for neonatal HIBD for full-term infants with moderate to severe HIE [10]. By this treatment, infants with the most severe type of HIE might not be saved, and this strategy involves the risk of severe disability or death, as nearly 50% of treated cases still die [11], and as many as 1/3 to 1/2 of sufferers demonstrate low Intelligence Quotient (IQ) at 6 to 7 years of age or constant neurologic abnormalities [12,13]. Therefore, more effective neuroprotective strategies are required. In this respect, several agents have been investigated to save the brain from irreversible damage or to postpone the pathological process, such as Erythropoietin, Melatonin, Metformin, stem cell therapy, etc., but only a few agents have been used in clinical studies [14]. Therefore, a search to explore better neuroprotective agents is under way.

In recent times, focus on plant research has increased globally and significant attention has been paid to medicinal plants due to their safety, cost-effectiveness, availability, long-term use, and limited side-effects. So far, the latest investigations have been looking at the potential of natural compounds isolated from plants, vegetables, fruits, and beverages to treat neurological disorders. Among them, the majority of naturally derived neuroprotective agents are polyphenolic compounds such as flavonoids which show protective effects in several animal models of experimental neurological disorders [15,16]. These naturally derived compounds have been reported as neuro-functional regulatory, anti-apoptotic, anti-inflammatory, and anti-oxidative agents [17,18,19]. In these topics, several plant extracts and plant-derived compounds, such as grape seed extract, resveratrol, and cannabinol, have been scientifically identified to be useful in preventing or treating animal models of neonatal HIBD.

Although in previous review articles, natural products were studied as neuroprotective agents in various neurological and neurodegenerative diseases, none of them have focused on neonatal HIBD. In addition to introducing neonatal HI brain damage and its pathophysiology, in this study, we aimed to review the literature in scientific databases that investigate the protective effects of plant extracts or their constituents in experimental animal models of neonatal HI brain damage and their possible underlying mechanisms of action. For this purpose, articles indexed in PubMed, Science direct, and Scopus databases were evaluated, using the following keywords in English: (Neonatal [Title]) OR (Perinatal [Title]) OR (Neonate [Title]) OR (Newborn [Title]) AND (hypoxic-ischemic [Title]) OR (hypoxia-ischemia [Title]) OR (hypoxic ischemic encephalopathy [Title]) AND (brain [Title]) AND (natural [Title/Abstract]) OR (plant [Title/Abstract]) OR (extract [Title/Abstract]) OR (herb [Title/Abstract]). In addition, the reference listing of the reclaimed essays and further resources comprising Google Scholar was screened.

## 2. Pathophysiology of Neonatal Hypoxic-Ischemic Brain Damage

### 2.1. Excitatory

In fact, HIE is caused by a restriction in blood supply (ischemia) and deprivation of adequate oxygen supply (hypoxia), which causes a switch to anaerobic metabolism. Anaerobic metabolism leads to depletion of adenosine triphosphate (ATP) and accumulation of lactic acid. This condition is known as primary energy failure, and occurs immediately after initial HI insult. At the cellular level, energy failure and ATP reduction results in failure of the cell membrane ion in pumping and consequently in flow of sodium and calcium ions accompanied by water flow into the cell. Following the process, the cytotoxic edema is eventuated due to the water flow which might lead to cellular swelling and cell lysis [9]. The following membrane depolarization unrolls voltage-sensitive calcium channels and leads to excessive calcium influx. Increase in intracellular Ca^2+^ leads to production and release of excitatory amino acids, particularly glutamate. In the following stage, excess glutamate activates receptors such as α-amino-3-hydroxy-5-methyl-4-isoxazolepropionic acid (AMPA) and N-methyl-D-aspartate (NMDA) receptors [20]. Activation of these receptors additionally promotes influx of Ca^2+^ into the affected cells. Finally, the rise in intracellular Ca^2+^ promotes production of reactive oxygen species (ROS), release of pro-radicals (such as free ions), and synthesis of excessive nitric oxide (NO) by neuronal NO synthase (nNOS), which is modulated by the NMDA receptors [20]. Furthermore, activation of lipases, proteases, and endonucleases are increased, which causes a release of fatty acids and membrane damage, and triggers a cascade reaction series which leads to cell death [20].

### 2.2. Free Radical Toxicity

Reoxygenation and reperfusion oxygen availability lead to the initial ischemic tissue and more ROS generation by mitochondria (via the electron transport chain) and damage integrity of membrane and organelle’s function. These activities lead to mitochondrial dysfunction which in turn can trigger ROS-induced ROS generation. Furthermore, hypoxanthine and pro-radicals such as iron cause large amounts of ROS formation. Mitochondrial impairment is one of the main factors involved in the apoptotic pathway of cell death. The second phase of injury is known as reperfusion injury and occurs hours later [20].

### 2.3. Inflammation

In addition to the described mechanisms, damaged neurons and activated endothelium produce various cytokines, including interleukins (IL-1, IL-6, IL-8, IL-10) and tumor necrosis factor (TNF)-α, which activate inflammatory response and trigger biochemical pathways that lead to secondary energy failure and a delay in neuronal cell death [20]. Secondary energy failure generally eventuates 6 to 48 h after initial HI insult and delays necrotic and apoptotic processes, that might continue up to days or weeks after birth [20]. Also, a delay in brain damage is associated with a decrease in production of neurotrophic growth factors, including epidermal growth factor (EGF), nerve growth factor (NGF), insulin-like growth factor (IGF), GDNF (glial cell-derived neurotrophic factor), BDNF (brain-derived neurotrophic factor), and VEGF (vascular endothelial growth factor), which apparently inhibit apoptosis and prompt cell proliferation and differentiation of the developing brain. On the other side, transcription factors comprising JNK (c-Jun N-terminal kinase) and NF-kβ (nuclear factor kappa-beta) also demonstrate an important function in this point [20]. The damage progression proceeds and enters into the tertiary phase by progress in inflammation, alteration in neurogenesis, and impaired synaptogenesis and axonal growth [9].

### 2.4. Neuronal Death

Neuronal cell death induced by neonatal HI occurs either by necrosis or apoptosis. Necrotic cell death occurs immediately after the insult and is the prevailing pathway of cell death subsequent to irreversible or severe damage [9]. Apoptosis or programmed cell death planned apoptosis is a significant part of ultimate cell death. Apoptosis can last for days and even weeks after the initial insult. It was proposed that, in infants, apoptosis is possibly more critical in provoking cell death compared with necrosis [20].

## 3. Potential Mechanisms for Neuroprotective Strategies

As mentioned above, HIE consists of complex pathophysiologic mechanisms occurring in two energy failure stages. The first stage eventuates immediately after HI and the secondary stage eventuates 6–15 h after the initial insult. This delay offers a temporary therapeutic window for neuroprotective strategies, during which pharmacological interventions can be applied. To date, a broad range of propitious neuroprotective structures have been evaluated on experimental animal models of neonatal HIBD, but only a few agents have been used in clinical studies [14,21].

The pathophysiology of HIE enables multiple targets at different time points of the disease process. As an example, in the first stage, treatments are mostly concentrated on reduction of oxidative, apoptotic, and excitotoxic mediators of injury, whereas in the later stages, reduction of neurotrophic-evoked excitation features and inflammatory cytokines in the immature brain are targeted to contribute to regeneration of oligodendrocyte and neurons [22,23].

### 3.1. Reduction of Cellular Apoptosis and Oxidative Stress in HI

Apoptosis is categorized into two pathways, one is caspase-dependent and the second is caspase-independent, and both play an important task in the neonatal pathogenic process in HIBD. Therefore, apoptotic pathways are the main target for therapeutic interventions. Apaf-1, caspase, and Bcl-2 gene families demonstrate the major role in neuronal apoptosis determination. The Bcl-2 gene family includes pro-apoptotic (Bok, Bak, and Bax) and anti-apoptotic (Bcl-XL and Bcl-2) members that regulate cytochrome c translocation from mitochondria to cytoplasm, which is the initiator phase of the caspase-independent apoptosis. The mechanisms of release of pro-apoptotic Bcl-2 from the membrane of mitochondria might include the rupture of membrane of mitochondria, and anti-apoptotic members prevent this process. Caspases are intracellular proteases that function as initiators, and effectors of apoptosis function to activate other caspases’ cleavage [24]. Numerous agents have been evaluated to function on anti-apoptotic pathways, for example the proteins of Bcl-2 family become a prosperous object for drug targeting. Caspases are also one of the intriguing classes of proteins as a pharmacological target since caspase inhibitors are able to block apoptosis, for example, casp-3 inhibitors demonstrated neuroprotective activities in animal models of neonatal HIBD [25,26,27]. Furthermore, it has been recently demonstrated that autophagy (programmed cell death) reveals a significant efficacy in HIBD and neuroprotection, which can be obtained by regulating autophagy.

As discussed above, a large amount of free radical species, mainly ROS and RNS (reactive nitrogen species), comprising hydroxyl radical, superoxide, peroxynitrite, and hydrogen peroxide, accumulate rapidly in the brain after HI and cause oxidative stress that is able to initiate protein oxidation, lipid peroxidation, nucleic acid, and cell membrane damage, and eventually, lead to cell death. Due to the high level of polyunsaturated fatty acids, low concentration of endogenous antioxidant enzymes: SOD (superoxide dismutase), CAT (catalase), GPx (gluthathione peroxidase), GST (glutathione S-transferase), GSH-Px (gluthathione peroxidase), and high oxygen consumption, neonatal encephalon is more sensitive to oxidative damage than adults. Oxidative stress has a critical role in neuroinflammation and apoptosis, especially in neonatal HIBD. In addition, ROS are in charge of the numerous downstream factors which directly change the blood–brain barrier (BBB) integrity, and also effect the tight junction (TJ) proteins’ modification and the matrix metalloproteinases’ (MMPs) activation. Thus, the activation of MMPs with regards to oxidative stress is deemed as an inappropriate factor to segregate the integrity of the BBB. According to the evidence, it has been demonstrated that the changes in permeability of BBB and oxidative stress are generally involved in the neonatal pathogenesis of encephalon injury following HI insult. Many agents with anti-oxidative properties, including free radical scavengers (FRS) or free radical production inhibitors, such as melatonin or erythropoietin, have been reported to provide a protective effect in animal models of neonatal HIE [28,29]. Furthermore, stabilization of the BBB after HIBD in neonates is a fundamental target in order to keep the microenvironment of cerebral homeostatic and the normal function of the neurons unchanged.

### 3.2. Reduction of Neurotrophic-Evoked Excitation and Inflammatory Cytokines in HI

Neuroinflammation subsequently occurs following HI, and anti-inflammatory treatments have been shown to heal brain injuries in neonates. For example, agents with anti-inflammatory effects such as allopurinol, deferoxamine, N-acetylcysteine, melatonin, and minocycline have demonstrated beneficial effects in animal models. Moreover, NF-kB inhibitors have also shown sustainable neuroprotection [30,31,32].

Supplementation of insufficient growth factors seems to be a promising therapeutic strategy, and administration of BDNF, IGF-1, and Erythropoietin ameliorate functional recovery and improve neurogenesis after neonatal HI injury. More recently, promoting neuronal regeneration with endogenous stem cells, such as mesenchymal progenitor/stem cells, has also been represented to promote functional outcomes in neonates [33,34].

### 3.3. Animal Model of HI

Animal models are normally the primary step in identifying mechanisms underlying illness and disease, to discover effective therapeutic treatments. Neonatal brain damage requires gestation for some months, and also the fetal and newborn maturation of the brain. Many animal models have reported information concerning the brain damage pathophysiology in term infants [35]. A combination of proper animal models to direct these investigations will help us to comprehend human neonatal damage and the approach of therapeutic interventions. In this review, the bulk of the studies reviewed three kinds of animal models, mice, rat, and piglet, in HI. Neonatal HI mice studies were performed as outlined in the following procedure: At postnatal day 7, pups were anesthetized, and the left common carotid artery was subjected and Bovie cauterized. Subjecting to hypoxia was carried out in sealed chambers immersed in a water bath [36]. In the rat model, usually, the neonatal rats (7 days old) were exposed to HIBD by the RiceVannucci modeling method. After anesthetizing rats, a longitudinal incision was made in the middle of their neck. The left common carotid artery was separated and was ligated on both sides. After cutting the blood vessel, the wound was quickly sutured [37]. In the piglet model, 1- to 2-day-old male piglets were anesthetized. Then, the two common carotid arteries were subjected, and they were tightened by elastic bands. After the surgery, HI brain injury was induced in the piglets by total interruption of the carotid blood flow by tightening the elastic bands around the arteries [38].

## 4. Pharmacological Evidence of Natural Plant Products in Neonatal Hypoxic-Ischemic Brain Damage

### 4.1. Plant Extracts

#### 4.1.1. Grape Seed Extract

Grape is one of the fruits that is a rich source of phenolic compounds and 60–70% of grape phenolic compounds are discovered in the seeds of the fruit. Grape seed extract (GSE) comprises several phenolic compounds, mainly procyanidins and proanthocyanodins monomers, polymers, and their gallate ester and resveratrol, which is the major compound that is extracted from the skin and seeds of grape [39,40,41]. GSE is a powerful Free Radical Scavenger (FRS) of oxygen and has anti-lipid peroxidation activity [32,33,34,35,36,37,38,39,40,41,42,43,44,45]. GSE revealed anti-oxidant efficacy more than vitamins C and E [46]. GSE also displayed anti-inflammatory and anti-apoptotic actions [47,48,49,50]. In addition, GSE can reduce brain injury in forebrain ischemia in a gerbil model in adults [51].

Zheng and coworkers, in 2005, investigated the neuroprotective effect of GSE against neonatal HI brain injury [52]. In this study, rat pups at postnatal day 7 (P7) were subjected to HI insult and received GSE (50 mg/kg by intra peritoneal (i.p.)) 5 min before hypoxia and 4 h after reoxygenation (twice daily for 1 day). Brain weight loss was reduced from 20.0% in vehicle rats to 3.1% in treated rats, as well as improvement in the histopathologic brain score in hippocampus, thalamus, and cortex after GSE pretreatment. Additionally, lipid peroxidation markers 8-isoPGF_2α_ (8-isoprostaglandin F2α) and TBARS (thiobarbituric acid reactive substances) levels significantly reduced. In another study [53], GSE was injected at 5 min to 5 h after reoxygenation and it was found that GSE post-treatment reduces neurofunctional abnormalities and brain weight loss even when administered 3 h after injury. In addition, the reduction of lipid peroxidation marker 8-isoPGF_2α_ and pro-apoptotic protein c-jun in the rat brain cortex suggested that GSE exhibits neuroprotection through free radical inhibition and anti-apoptotic effects [53].

#### 4.1.2. Grape Seed Proanthocyanidin Extract

Grape seed proanthocyanidin extract (GSPE) is a combination of biologically active flavanol compounds ranging from monomers such as gallic acid, epicatechin, catechins, and their gallate forms, to oligomeric proanthocyanidins. GSPE is known as one of the more potent natural antioxidants and its antioxidant capacity is significantly higher than vitamins E and C [54]. GSPE has been identified in various plants, for instance, cinnamon bark, pine bark, lotus, and apple, and also is widely distributed in red and white grape seeds. Many studies have reported a variety of pharmacological activities for GSPE including antioxidative [55,56], anti-inflammatory, anti-apoptosis [43], and anticancer properties. Recent studies have also been reported that suggest that GSPE possesses neuroprotective effects [56,57,58], for example against an adult rat model of ischemia-reperfusion injury [59].

Tu et al. demonstrated that GSPE significantly decreased brain infarct volume (approximately 50%) and improved neurobehavioral recovery in HIE mice models when pups were pretreated with GSPE (30 mg/kg, intraperitoneal (i.p.) injection) 20 min before HI [60]. In addition, the number of apoptotic cells and pro-apoptotic proteins’ expression (cleaved caspase-3 and bax) was significantly reduced, while expression of anti-apoptosis protein bcl-2 and the bcl2/bax ratio increased in the pretreated group, demonstrating the antiapoptotic role of GSPE [60].

#### 4.1.3. Pomegranate Juice and Pomegranate Polyphenol Extract

The scientific name for pomegranate is *Punica granatum* Linn. Different parts of pomegranate (peel, seeds, and arils) are an important source of polyphenols, including tannins, mainly ellagitannins and gallotanins, flavonoids, mainly luteolin, quercetin, anthocyanidins, catechin, and epicatechin, alkaloids, etc. Moreover, pomegranate contains hydroxybenzoic acids such as gallagic acid, ellagic acid, and ellagic acid glycosides [61,62,63,64]. Many studies have demonstrated that pomegranate has been linked with the prevention and treatment of a wide range of disorders, such as cardiovascular disease [65], hypertension [66], cancer (skin, breast, prostate, colon, lung, pancreatic) [67,68], arthritis [69,70], hyperlipidemia [71], obesity [72], diabetes [73], and wound healing activities [74]. The therapeutic potential of pomegranate fractions are correlated with its antioxidative [75,76,77], anti-inflammatory [78,79], anti-infective [80], anti-atherogenic, anti-hyperglycemic, immunomodulation [81,82], and anti-proliferative [83,84] properties. In recent decades, a number of in vivo and in vitro experimental researches have indicated the neuroprotective properties of pomegranate [85,86,87] against neurodegenerative diseases, particularly Alzheimer’s and depression-related models in several animal models [88,89,90]. In addition, other studies have demonstrated the effect of pomegranate and pomegranate seed extract on improvement of memory deficit in rat models of cerebral ischemia [91,92,93] and ischemia-induced anxiety in male rats [94]. In another study, pomegranate has been shown to improve cognitive and functional recovery after ischemic stroke [95].

In order to investigate the neuroprotective effect of pomegranate juice against neonatal HI brain insult, in 2005, Loren et al. [36] conducted a study on mature C57BL6 mice. In this study, pregnant mice received pomegranate juice in their drinking water at three dose levels (1:60, 1:80, and 1:320 dilutions of pomegranate juice concentrate). The treatment time for total groups was, in most cases, 15 days, which included 7 days in utero and 8 days ex utero for mice that sustained caspase-3 analysis, and 21 days, which included 7 days in utero and 14 days ex utero for mice that sustained histologic analysis. Seven-day-old neonatal mice were subjected to hypoxic-ischemic insult. They found significantly (>60%) decreased brain tissue loss as well as a decrease in activation of caspase-3 (84% in the hippocampus tissue and 64% in the cortex tissue) in the treated group as compared to control pups. The results demonstrated the effectiveness of pomegranate juice against neonatal HIBD through maternal dietary supplementation. The neuroprotective effects were observed. In a follow-up study, using the same neonatal HI mouse model, Loren et al. showed that maternal supplementation with pomegranate polyphenol extract (PPE) resulted in a significant decrease of HI-induced caspase-3 activation. The authors suggested that phenolic compounds of the juice are responsible for its neuroprotective effect [96]. Recent studies demonstrated that among polyphenol-containing foods, pomegranate juice had the highest concentration of measurable polyphenol due to the fact that the pomegranate juice antioxidant capacity was higher than green tea or red wine. Anthocyanins are the largest and most important group of flavonoids which exist in pomegranate juice [61,62,97,98].

#### 4.1.4. *Dendrobium officinale* Extract

*Dendrobium officinale* (DO), that is used in traditional Chinese herbal medicine, represents various beneficial therapeutic activities, like chronic inflammation reduction. The neuroprotective activity of DO extract in ischemia-reperfusion-induced arrhythmia rat models has also been reported [99,100,101].

In 2019, Li found that DO extract (DOE) exerted neuroprotection against neonatal HIBD when 7-day-old rats were intra-gastrically administered different doses of DOE ((75–150–300) mg/kg/day, continuously for 14 consecutive days) after HI [37]. Results showed that after treatment with DOE, the levels of MDA (malondialdehyde), NO (nitric oxide), and NOS diminished and SOD activity increased in a dose-dependent manner, which indicated the enhancement of the antioxidant capacity in HIBD rats. In addition, the behavioral ability of rats significantly improved after DOE treatment in a dose-dependent manner. Besides, DOE led to downregulation of Bax and caspase-3 and upregulation of Bcl-2 (demonstrating reduced neuronal apoptosis). Moreover, CI area percentage was remarkably diminished after treatment. They also examined the protein expression related to brain injury (KCC2 and HDAC1) and it was reported that after treatment with DOE, the expression of HDAC1 was diminished and KCC2 expression was elevated in the hippocampus and cortex. Meanwhile, determining the neurotrophic factors’ expression: HIF-1α (hypoxia-inducible factor 1- alpha), bFGF (basic fibroblast growth factor), CNTF (ciliary neurotrophic factor), and BDNF, after treatment with DOE, indicated that HIF-1α expression was remarkably diminished, and the bFGF, CNTF, and BDNF expression were enhanced. Altogether, this study demonstrated that the DOE is able to increase the neurotrophic factors’ expression and leads to neuronal apoptosis suppression and stimulates antioxidant capacity.

### 4.2. Phytochemicals

#### 4.2.1. Verbascoside

Phenylethanoid glycosides are a kind of water-soluble natural phenolic compound that have exhibited neuroprotective activity [102]. Verbascoside (VB) (the chemical structure is shown in Figure 1a) is a typical phenylethanoid glycoside from lemon verbena [103]. VB is found in more than 150 types of plants [104]. There is numerous evidence that VB has various biological activities, including antioxidant [105,106], anti-inflammatory [107,108,109], immunoregulatory [110], anticancer [111], and antimicrobial. In addition, several research studies have shown that VB also has neuroprotective properties against various neurological disorders in in vitro and in vivo experiments [112,113,114].

Wei et al. [115] investigated the neuroprotective effects of VB on a HIBD model in rat neonates. In this study, animals were subjected to HIBD at p7 and VB groups (60, 120, and 240 mg/kg) were administered through i.p. injection every 12 h after HI for two consecutive days. VB post-treatment (240 mg/kg) significantly reduced prolonged reflex latencies and brain infarct volume dose-dependently (120 and 240 mg/kg). The VB-treated group (240 mg/kg) also remarkably decreased the degree of degeneration, morphological damage, and necrosis in the cortex and hippocampus CA3 region compared with the HI group. Furthermore, autophagosome formation and the autophagy-related proteins’ expression (P62, Beclin-1, and LC3-II/I ratio) reduced after VB post-treatment.

#### 4.2.2. Geniposide

Iridoid glycosides are phytochemicals that naturally occur in many plants [116,117]. Geniposide (Figure 1b) is one of the major iridoid glycoside compounds that are purified from the Chinese herb *Gardenia jasminoides* [118]. Nearly 40 plants have been identified that contain geniposide [119]. Geniposide possesses diverse pharmacological activities, including anti-inflammatory [120,121], antidiabetic [122], anti-oxidative, and hepatoprotective [123]. In recent years, geniposide demonstrated excellent neuroprotective activities in experimental neurological dysfunctions, such as Parkinson’s disease [124], Alzheimer’s disease [125,126], and cerebral ischemia [127,128,129].

Liu et al., in 2019, studied the therapeutic effects and underlying mechanisms of geniposide against HI brain damage in neonatal mice. In this study, mice pups were subjected to HI insult at P10 and geniposide was administered (20 mg/kg) daily, intra-gastrically after HI. Findings showed that geniposide treatment significantly reduced the number of apoptotic neurons and also decreased the leakage of serum Immunoglobulin type G (IgG) into brain tissue. In addition, mRNA expression levels of pericyte markers, adherens, and tight junction proteins were upregulated after treatment, which meant that geniposide decreased the disruption of the BBB, which was induced by HI. Moreover, geniposide post-treatment attenuated astrogliosis and microgliosis. Microglia and astrocytes are two of the major neuroinflammation mediators [130]. To further elucidate the underlying molecular mechanism of geniposide-induced neuroprotection, it was found that Phosphoinositide 3-kinases/Protein kinase B (PI3K/Akt) expression signaling pathway-related protein was increased after treatment, which showed that geniposide probably exploits its neuroprotection activity via PI3K/Akt signaling pathway activation.

#### 4.2.3. Hesperidin

Hesperidin (Figure 1c) belongs to the flavanone glycoside class of compounds, which was originally discovered in citrus fruits [131] but has been reported to occur in many plants other than citrus [132]. Several biological activities have been reported from Hesperidin, for example, antioxidative [133,134], anti-inflammatory [135], anticarcinogenic, and antiallergic properties. Hesperidin also has the ability to pass through the BBB [136]. In recent studies, both the in vivo and in vitro neuroprotective properties of hesperidin have been shown [137], which are attributed to its antioxidant and anti-inflammatory activities. This compound also had neuroprotective effects on amyloid β [138] against 3-nitropropionic acid-induced [139,140] and H_2_O_2_-induced [141] neurotoxicity. It also possesses antidepressant activities [142]. Another study reported that hesperidin enhances learning and memory in a rat model of cerebral ischemic-reperfusion injury [143].

Rong et al. demonstrated that hesperidin pretreatment in HIBD rat neonates increased the surviving brain volume from 49.8% to 72.9% and improved long-term behavioral development in oral administration of hesperidin (50 mg/kg/day) during 3 days in animals. Hesperidin pretreatment also markedly decreased Fluoro-Jade B stain-positive neurons in the cortex of the brain at post-HI [144]. In the culture media, pretreatment with hesperidin (1.6 μM) reduced the release of LDH (lactate dehydrogenase) and enhanced the levels of 3-(4,5-dimethylthiazol-2-yl)-2,5-diphenyltetrazolium bromide (MTT), which indicated that hesperidin promotes neuronal survival. Moreover, levels of ROS and MDA decreased with hesperidin pretreatment in both in vivo and in vitro experimental models. Results also showed that hesperidin pretreatment suppressed FoxO_3_ phosphorylation, which indicated that possible molecular mechanisms of hesperidin neuroprotection are likely the result of activation of the PI3K/Akt survival signaling pathway and free radical reduction.

#### 4.2.4. Quercetin

Quercetin (Que) is a flavonoid (3,3′,4′,5,6-Pentahydroxyflavone) (Figure 1d) which is distributed all over the world in fruits, vegetables, and many other dietary sources, including green tea, red wine, berries, and onions [145]. Quercetin is generally known as a strong antioxidant and free radical scavenger [146,147,148]. It also possesses anti-inflammatory and neuroprotective effects in brain injuries, such as traumatic [149] and ischemic/reperfusion [150]. Many recent studies have been conducted to examine the neuroprotective effects of quercetin on ischemic brain damage [151] in vitro [152,153] and in various types of in vivo models, such as focal cerebral ischemia/reperfusion [154,155] and global cerebral ischemia [156]. Moreover, in some studies, delivery systems such as liposomal [157] and the nano-encapsulation approach have been used to deliver quercetin into the rat’s brain [158].

Qu, in 2014, conducted a study on postnatal 3-day-old rats and found that administration of Que (intra-gastrically once a day, 20 or 40 mg/kg from 2 h after HI to the day rats were sacrificed) significantly improved HI-induced myelin damage through strengthening survival of oligodendrocytes [159]. In 2019, Wu used a 7-day-old neonatal rat model of HIBD and administered Que (40 mg/kg/day) intra-gastrically once a day for 7 days. Results showed that Que treatment enhanced anti-apoptotic (Bcl-2) and reduced pro-apoptotic (Bax) proteins in cortical cells, as well as significantly attenuated the DNA-strand break induced by HI, which demonstrated the antiapoptotic effect of Que. Data also showed that Que treatment significantly decreased astrogliosis and microgliosis and downregulated the inflammatory factors’ expression (TNF-α, IL-1β, and IL-6) in rat cortex, which showed that Que hads an anti-inflammatory effect and could protect brain cortex tissue from subsequent damage [160]. To study the mechanism of Que, authors examined the expressions of the TLR4/NF-κB signaling pathway. Based on the results, the phosphorylation of TLR4 (Toll-like receptor 4) reduced significantly and signals of p-p65 and p-p-IκBα were downregulated. These data confirmed that Que probably exerts its neuroprotection activity via suppression of the TLR4-mediated NF-κB pathway.

#### 4.2.5. Pterostilbene

Pterostilbene (PTE) (3,5-dimethoxy-4-hydroxystilbene) (Figure 1e) is one of the resveratrol derivatives and was predominantly discovered in numerous varieties of grapes and blueberries [161,162]. PTE belongs to the phytoalexins family, and biosynthesizes in plants in response to biotic and abiotic stress [163]. The following evidence demonstrates that PTE possesses a protective effect against oxidative stress, inflammation, apoptosis, and shows anticancer and analgesic activities [164,165,166]. Recent studies suggest that pterostilbene also has neuroprotective effects in nervous system disorders such as lipopolysaccharide-induced learning and memory deficit [167], Alzheimer’s disease [168], anxiety [169], and ischemia-reperfusion brain damage [170,171].

In 2016, Li and coworkers reported that PTE (single dose—50 mg/kg) had neuroprotective effects against HI brain damage in rats when injected i.p. 30 min before insult. Brain infarct volume and brain edema markedly decreased (nearly 50%) and animals’ survival enhanced (from 78% to 95%). Additionally, PTE pretreatment significantly decreased neurological score and improved working memory impairment and motor deficit. Moreover, pro-inflammatory cytokines’ expression (IL-1, IL-6, and TNFα,) and transcription factors (p-65 and NF-κβ) decreased in hippocampus CA1 and represented an anti-inflammatory effect of PTE. The results also showed inhibition of programmed cell death. Moreover, the level of TBARS and ROS were decreased and SOD activity was increased, which indicated the antiapoptotic and antioxidative role of PTE [172].

PTE pretreatment significantly prevented mRNA and protein expression of HO-1 (hemeoxygenase-1). On the other hand, HO-1 inhibitor, ZnPP, can inhibit the neuroprotective effect of PTE. Authors suggested that HO-1 is a possible target of PTE in neuroprotection for oxidative, apoptotic, and inflammatory damage induced by HI [172].

#### 4.2.6. Vitexin

Vitexin (5, 7, 4-trihydroxyflavone-8-glucoside) is an apigenin flavone glycoside (Figure 1f) that has been discovered in several plant species, including *Vitex negundo* seed [173], Passion flower [174], mimosa [175], wheat leaves [176], and hawthorn [177]. Earlier studies revealed that it possesses potent pharmacological actions such as antioxidant, anticancer [178], antiviral [179], anti-inflammatory [180], antihypertensive [181], and anti-depressant-like actions [182]. In addition, there are few reports on the neuroprotective efficacy of vitexin [183,184,185] such as its protective effect against cerebral I/R (ischemia/reperfusion) [186]. Vitexin is an HIF-1α inhibitor [187] and former researches have represented that the early suppression of HIF-1α after HI injury produced neuroprotection.

A study by Min et al. [188] revealed that administration of vitexin (single-dose: 30–60 mg/kg i.p.) at 5 min or 3 h after HI can significantly decrease brain infarct volume and improve long-term behavioral development as well as reduce neuronal cell death and brain tissue loss in neonatal rats dose-dependently. In addition, vitexin significantly inhibited upregulation of HIF-1α and VEGF protein levels. Vitexin also reduced the BBB disruption following brain edema. Throughout hypoxia, HIF-1α showed a significant function in VEGF expression. VEGF had a destructive effect on the surrounding vasculature that resulted in BBB disruption and subsequently, brain edema. Previous reports have shown that the VEGF expression inhibition throughout the primary cerebral ischemia stage or hypoxia led to the BBB protection. These data suggest that the primary HIF- 1α inhibition along with vitexin decreased VEGF levels and led to protection of the BBB and brain edema reduction. Additionally, vitexin (45 mg/kg) was the most efficacious dose at 5 min after HI but lost its effects when administered 3 h after HI. These results indicated that the neuroprotection activity of vitexin is related to primary inhibition of HIF-1α and VEGF expression.

#### 4.2.7. Oxymatrine

Oxymatrine (OMT) (Figure 1g) is one of the main alkaloids isolated from traditional Chinese herbal medicine *Sophora japonica* [189]. Oxymatrine has indicated anti-inflammatory [190,191], antiviral [192], antioxidant [193], anti-hepatic fibrosis [194], anti-apoptosis [195,196], and anticancer [197,198] effects. In recent years, in vitro [199] and in vivo investigations have represented that OMT possesses a neuroprotective effect against experimental brain injury models such as traumatic injury [200] and ischemia/reperfusion damage [201,202].

Liu et al., in 2015 [203], reported that OMT could ameliorate brain damage in HI rat neonates. In order to identify OMT’s specific effect on HI rat neonates and underlying mechanism, another in vivo and in vitro experimental study was designed on a model of neonatal HIBD [204]. In this study, 7-day-old rat pups were administered oxymatrine (120 mg/kg, i.p. injection) after HI every 12 h for 2 days. OMT treatment attenuated neuronal damage and necrotic cell loss and degeneration in cerebral hippocampus CA3 and cortex. In addition, the reflex latencies were reduced in treated pups.

The results of cultural hippocampal neurons obviously indicated that OMT improved cell viability, reduced LDH leakage, and attenuated intracellular Ca^2+^ and loss of MMP, and decreased the rate of cell apoptosis in neurons exposed to OGD/RP. In addition, OMT adjusted the expression of the mRNA and protein of the PI3K/Akt/GSK3β pathway, inhibited the expression of NR2B protein, and regulated apoptosis mediators (reduced caspase-3 and enhanced Bcl-2 and MCL-1 levels). These results demonstrated that OMT exerts neuroprotection against neonatal HIBD via regulating apoptosis, NR2B downregulation, and activation of the PI3K/Akt/GSK3β signaling pathway [203,204].

#### 4.2.8. Coumestrol

Neuroprotective effects of estrogens against ischemic brain damage have been demonstrated previously [205]. Phytoestrogens are naturally occurring estrogen-like molecules found in many dietary plant sources and are structurally similar to estradiol [206,207,208]. Isoflavonoids are a subgroup of phytoestrogens with estrogen agonist ability and mimic estrogen’s neuroprotective properties [209,210]. Coumestrol (Figure 1h) is an isoflavonoid-like natural compound present in soybean, alfalfa, and red clover [211,212,213], and various studies have proven its anti-inflammatory [214], anti-adipogenesis [215], anti-aging [216], antioxidant, and anticancer [217,218,219] activities. In addition, the literature also indicates that coumestrol exerts neuroprotection against Alzheimer’s disease [220,221], multiple sclerosis, and brain global ischemia [222,223].

Recently, in one study, effects of coumestrol administration on neonatal HIBD have been investigated in male rats [224]. In this study, coumestrol (20 mg/kg) was injected i.p. immediately before HI, as a preventive, or 3 h post-hypoxia as a therapeutic intervention. Results showed that coumestrol pretreatment prevents HI-induced mitochondrial swelling. Furthermore, pre- and post-treatment with coumestrol protected the reference and the working memory from impairments as well as partially prevented late cognitive deficits in both the reference and working spatial memory tasks that were evaluated at P60. In addition, pre- or post-hypoxia treatment reduced tissue death in both ipsilateral brain hemisphere (20% recovery) and hippocampus (25% recovery). Finally, Coumestrol pre- and post-treatment attenuated late GFAP (glial fibrillary acidic protein, a marker of late HI-induced reactive astrogliosis) overexpression in the hippocampus and also demonstrated that this compound can provide long-term protective activity for astroglia against cell damage. Based on these data, the authors suggested that coumestrol neuroprotection mechanisms involve prevention of mitochondrial dysfunction and long-term cognitive deficits.

#### 4.2.9. Plastoquinone

During the photosynthesis process, a large amount of ROS is generated in plants by the sunlight. So, chloroplasts are kept by a potent antioxidative system. Plastoquinone (Figure 1i) is an important component of the photosynthetic electron transport chains in chloroplasts and the reduced form of plastoquinone (plastoquinol) is a powerful antioxidant which diminishes generated ROS during photosynthetic reactions [225,226,227,228].

As mentioned above, HI causes mitochondrial dysfunction which actuates the pathological ROS production. Thus, antioxidants’ targeted delivery to the mitochondria is a productive therapeutic strategy to treat the detrimental impacts of HIBD. Previous data have indicated that SkQR1 (10-(6′-plastoquinonyl) decylrhodamine 19) (mitochondria-targeted antioxidant), which is a plastoquinone molecule linked to rhodamine derivative [229], has a protective effect on ROS-induced pathological conditions in a number of tissues, such as heart [230], kidney [231,232], eye [233], and brain [230,231,232,234,235]. SkQR1 also showed a neuroprotective effect in experimental models of brain trauma [232] and brain ischemia [236].

The neuroprotective effect of SkQR1 in neonatal HIBD was explored by Silachev et al. in 2019 [237]. They found that pretreatment with SkQR1 in neonatal HI rats at a dose of 2 µmol/kg (i.p. injection) markedly reduced infarct volume and diminished oxidative stress in the brain, as well as reduced nervous tissue loss. SkQR1 pretreatment also led to a significant decrease of long-term neurological deficits. Finally, SkQR1 improved short- and long-term behavioral deficiency as a consequence of HI-induced and remarkably enhanced sensorimotor performance. Administration of SkQR1 before or immediately after (10 min) HI exerts neuroprotection via diminishing brain deterioration and ameliorating long-term neurological tasks. It seems that the rationale of neuroprotective action of SkQR1 can reduce the ROS generation with mitochondrial origin.

#### 4.2.10. Resveratrol

Resveratrol (3,5,4′-trans-trihydroxystilbene) (RES) is a natural phenolic compound with a stilbene structure (Figure 1j). Resveratrol is found naturally in various plant species; however, its dietary sources are limited. Some dietary sources of RES are peanuts, pistachio, berries, plums, pomegranates, grapes, and red wine (grapevines). Of all of these, grapes and red wine present the highest content [238,239,240]. RES has received extensive attention for its diverse pharmacological activities [241,242,243,244,245,246]. Besides, accumulating evidence indicated that RES treatment exerted neuroprotection against various brain disorders [247,248,249], including neurodegenerative diseases [250,251,252] such as Parkinson’s, Alzheimer’s, and Huntington’s diseases [253,254]. In addition, resveratrol also has neuroprotective effects in animal models of stroke [255], traumatic brain injury [256], and various kinds of ischemic brain injury ischemia [257,258] such as focal cerebral ischemia [259,260,261], global cerebral ischemia [262,263], and ischemia/reperfusion [264,265,266,267]. The RES mechanisms of action in neuroprotection might comprise antioxidation [268,269], anti-inflammation [270,271], and anti-apoptosis [272] effects.

Few reports have exhibited potential neuroprotective effects of RES in the pilot model of neonatal HI encephalopathy using both in vitro and in vivo approaches. West et al., in 2007, investigated the resveratrol effects on HI brain injury in both neonatal rats and mice [96]. In this study, the pups subjected to HI at postnatal day 7 and different concentrations of RES (20 mg/kg, 200 μg/kg, and 2 μg/kg) at several different points in time (24 h before hypoxia, 10 min before hypoxia, and 3 h after hypoxia) were administered via i.p. injection. Results indicated that RES at doses of 200 μg/kg or greater diminished caspase-3 and calpain activation at 24 h after the injury time dose-dependently, but only when its use before the injury showed both anti-apoptotic and necrotic cell death activity. RES also protected the brain from tissue loss 7 days after the injury. In addition to its protective effect in mouse neonates, RES also protects neonatal rats against HI, interestingly, when it is used after the injury. In another study in 2008, rat pups received RES (30 mg/kg) i.p. 30 min before or 30 min after HI. Results displayed an anti-apoptotic mechanism of RES via a remarkable decrease in expression level of Bax, the ratio of Bax/Bcl-2, and caspase-3, upon administration of RES at 30 min before HI but not when it was used 30 min after the injury [273]. Most of the surveys highlighted the short-term immunohistochemical and histological measurement of a set of biological markers and generally downgrade the long-term adverse effects and consequences of the treatment. In 2011, Karalis et al. investigated the delayed consequence of RES early administration through analysis of behavior and late neuropathological examination. Shortly, rats at P7 experienced HI and were treated instantly after hypoxia, receiving the 90 mg/kg of RES via i.p. injection. The remarkable difference was reported among the groups in righting water maze, rotarod, and reflex tests. According to the neuropathology study, a remarkable decrease in preservation of myelination and infarct eventuated after RES treatment. These results demonstrate that RES long-term neuroprotective activity on neonatal HI-induced damage on the gray and white tissue of the brain may be connected with the protection of behavioral functions [274].

Arteaga et al. conducted a study aiming to determine the RES activity as a preventive (before HI injury) or therapeutic agent (immediately after HI injury) on cellular and morphological damage and impairments in behavior in rat neonates. They found that pretreatment with RES reduced volume of infarction, preserved myelination, and minimized the astroglial reactive response. Moreover, long-term cognitive impairment significantly improved in adulthood. It was noticeable that none of these protective characteristics were found when RES was administered following HI [275]. All of these reported studies demonstrated the protective property of RES only when administered before HI (pre-treatment). However, Pan et al. reported that post-treatment with RES (100 mg/kg, i.p. injection three times at 0, 8, and 18 h respectively, after HI) remarkably decreased brain damage. They found that RES also decreased the expression of important inflammatory factors at the protein and mRNA levels, and at least partially through inhibition of microglia activation. However, RES demonstrated an anti-apoptotic activity, and changed the apoptosis-related genes’ expression, such as caspase3, Bax, and Bcl-2. These data indicate that inhibition of inflammation and apoptosis is a prosperous therapeutic strategy of RES post-treatment. The post-insult administration of RES has higher clinical relevance than pre-insult administration, though its specific molecular mechanism in the treatment of neuroinflammation is unclear [276]. Therefore, recently, Le and coworkers investigated the potential anti-inflammatory mechanism of RES on immature brains. They used the HIBI model in neonatal mice for in vivo and microglial cells for in vitro study. They found that RES significantly reduced the levels mRNA and protein, which were related to cytokines (TNF-α, IL-1β, and IL-6) in brain tissues following HI. As a result, RES remarkably ameliorated brain damage and neurobehavioral deficits by reducing neuroinflammatory responses. Additionally, RES reversed the reduction in the levels of SIRT1, like raising the levels of TLR4, MyD88, and NF-*κ*B, whereas the administration of SIRT1-specific inhibitor prior to RES significantly reversed this effect, which demonstrated that RES exerted anti-inflammatory activity via inhibition of the TLR4/MyD88/NF-*κ*B signaling pathway in vivo [277].

#### 4.2.11. Piceatannol

Piceatannol (PIC) (Figure 1k) is a polyphenolic stilbene phytochemical that is found in large amounts in passion fruit seeds, berries, and grapes [278,279]. PIC is a hydroxylated analog of RES but has stronger free-radical-scavenging activity as well as greater bioavailability than RES [280,281]. Previous studies have reported that PIC has antioxidant, anti-inflammatory, and anticancer activity [282,283]. In addition, it exerts cardioprotective and neuroprotective properties [284,285,286].

Dumont et al. [287] recently showed that nutritional supplementation of pregnant and breastfeeding female rats with PIC (0.15 mg/kg/day via drinking water) could provide neuroprotection for HI neonates. In behavioral terms, maternal supplementation of PIC restored sensorimotor functions and recovered cognitive functions. Also, the number of pups with brain lesions, brain lesion size, and severity of edema inside brain lesions decreased in the treated group. Moreover, the spatial distribution of white matter fiber bundles significantly improved. These results were observed in pups as early as 3 h post-injury, which indicated the short-term effect of PIC. In addition, the hippocampal-dependent memory of pups was completely restored at P30, which indicated the long-term neuroprotective effect of maternal PIC consumption. The PIC-induced neuroprotection could be attributed to its antioxidant effect or modulation of the brain metabolism.

#### 4.2.12. Polydatin

Polydatin (Figure 1l) is the main component of *Polygonum cuspidatum* (a traditional Chinese herb) but is also found in many daily diets, such as grape, peanut, hop cones, red wines, and chocolate. Polydatin is the glycoside of RES and the most abundant derivative of RES in nature [288]. Numerous studies have reported that polydatin has various pharmacological activities, comprising cardiovascular protection, anti-inflammation, antioxidation, anticancer, etc. [289]. Protective effects of polydatin against brain damage in animal models of ischemia-reperfusion have also been reported [290,291].

In one study by Sun et al., rat neonates were subjected to HIBI at P7 and polydatin (10 mg/kg) was administrated through i.p. injection once a day for 10 consecutive days. The results of a behavioral test showed that polydatin treatment led to enhancement of long-term learning and memory. In addition, BDNF expression was increased significantly in brains of treated pups. These results demonstrated that the protective effect of polydatin was possibly mediated via upregulation of BDNF, which showed that polydatin may have an important role in promoting neuronal survival and neuronal recovery following injury [292].

#### 4.2.13. Vanillin

Vanillin (Figure 1m), the major component of vanilla orchids, is a natural aromatic compound. Vanillin is one of the best flavor ingredient in cosmetic, food, and drug products [293]. It has been reported that vanillin exhibits various pharmacological activities [294]. Recent reports revealed that vanillin could pass through the BBB and showed neuroprotective effects. For instance, vanillin reduced ischemia-induced neuronal damage in gerbils [295] or it showed a protective role against Parkinson’s disease [296].

In one study, the authors examined neuroprotective effects of Vanillin post-treatment on HIBD in rat neonates at postnatal day 7. Evaluation of brain damage revealed that vanillin (80 mg/kg, i.p. injection) significantly improved HI-induced neurobehavioral deficits and reduced brain infract volume as well as brain edema in a dose-dependent manner. In addition, neuronal degeneration and necrotic cell death were reduced in the hippocampal CA1 and CA3 and cortex regions in the brains of the treated group. Furthermore, vanillin post-treatment decreased IgG leakage into the brain tissue, reduced the levels of MMPs, and upregulated the TJ-related proteins’ expression, which indicated that vanillin could improve BBB ultrastructure damage. Besides, endogenous antioxidant enzymes activities (SOD, GSH-Px, and CAT) and T-AOC (total antioxidant capacity) remarkably increased, and lipid peroxidation (reduction in the level of MDA content) was decreased. These results demonstrated that neuroprotection of vanillin might be attributed to its efficacy in oxidative stress inhibition and preserving BBB integrity, and reducing subsequent brain edema [297].

#### 4.2.14. Caffeine

Caffeine (Figure 1n) belongs to methylxanthine alkaloid, which is found naturally in many plants. Coffee and tea are the main sources of caffeine. The main action of the methylxanthines is to stimulate the central nervous system. Caffeine is a non-selective adenosine antagonist and shows a wide range of actions on the brain [298,299]. Caffeine has been widely used to treat apnea of prematurity for the past 30 years as well as to facilitate the extubation in very low birth weight infants. It is available in the form of caffeine citrate and its active substance is caffeine base [300,301,302].

In one study by Back et al. [303], neonatal mice were subjected to hypoxia and administered caffeine. It was found that hypomyelination and ventriculomegaly were diminished by caffeine treatment. Results also led to a more normally arranged myelinated axon orientation in the caffeine-treated group. Additionally, the proportion of immature oligodendrocytes was enhanced. In order to determine the underlying mechanism of caffeine’s neuroprotective effect, in another study, rat pups were subjected to HI at postnatal day 7 and received caffeine citrate (20 mg/kg) immediately before and at 0, 24, 48, and 72 h after HI [304]. Results demonstrated that administration of caffeine citrate led to a significant decrease in the number of apoptotic cells in the parietal cortex and hippocampus of the treated rat’s brain, which showed that the antiapoptotic effect of caffeine was possibly responsible for its protective effects against neonatal HI brain injury.

#### 4.2.15. Crocin

*Crocus sativus* L. stigma, generally known as saffron, is widely used to treat various diseases. Crocin (Figure 1o) is a water-soluble carotenoid and one of the main pharmacological active substances isolated from saffron. Previous studies have reported that crocin has high antioxidant capacity along with anti-inflammatory, anti-atherosclerotic, anticancer, and hypolipidemic effects [305]. Beside, neuroprotective properties of crocin have also been reported in experimental models of various brain disease, like cerebral ischemia-reperfusion [306,307], memory impairment [308], traumatic brain injury [309], and depression-related models [310].

Hypothermia therapy is one of the standard treatments for HIE in infants, although the efficacy is limited. Combination treatments are considered to increase the efficacy of hypothermia. Increasing the neuroprotective activity of hypothermia, combination treatments are deemed effective, and numerous therapeutic treatments have been examined as potential strategy in combination with hypothermia, comprising stem cells, erythropoietin, etc.

In one study [311], Huang and Jia investigated the protective effect of crocin and its combination with hypothermia against HIE in mouse neonates. Pups were subjected to HI at P7 followed by treatment with crocin (10 mg/kg) and hypothermia in combination. Levels of NO, ROS, and MDA were significantly reduced as well as mRNA expression of COX-2 (Cyclooxygenase-2) and iNOS (inducible nitric oxide synthase), which demonstrated the enhancement of the anti-oxidative effect of hypothermia after combined treatment. Furthermore, levels of IL-1β and TNF-α were markedly reduced, which indicated the enhancement of the anti-inflammatory effect of single hypothermia by combined treatment. Finally, combined treatment led to a significant reduction of neurological severity score, that showed the improvement of neurological function. These results demonstrated that crocin could increase the neuroprotective effect of hypothermia and can be used as a combined therapy with hypothermia.

#### 4.2.16. Tanshinone IIA

Danshen, isolated from the *Salvia miltiorrhizae* Bunge, is a popular traditional Chinese herbal medicine which has been used for the treatment of cardiovascular and cerebrovascular disease. Tanshinone IIA (Tan IIA) (Figure 1q) is the most abundant pharmacologically active component of Danshen [312,313,314]. Tan IIA possesses anti-oxidative, anti-inflammatory, anti-apoptosis, antiplatelet aggregation, and anticancer activities [315,316,317], and exerts protective effects against cerebral I/R injury [318]. Tan IIA can be defined as an AchE (acetylcholinesterase) inhibitor that has the ability to penetrate the BBB [319].

Xia et al. found that the HIBI induced in the P7 neonatal rat model with administration of tanshinone IIA (i.p. injection at 10 mg/kg/d from 2 days before HI for 9 or 16 days) improved neuropathology and sensorimotor functions. Also, Tan IIA remarkably enhanced the antioxidant capacity in rat plasma and protected the C17.2 progenitor cells from AAPH-induced death. Furthermore, Tan IIA exhibited a protective activity on mitochondrial membrane potential [320].

#### 4.2.17. Cannabidiol

Cannabinoids are a family of lipid mediators which modulate the synaptic transmission. Cannabinoids have emerged as promising neuroprotective agents in various brain disorders such as seizures, epilepsy, and Multiple Sclerosis (MS) [321,322,323]. In recent years, in vitro and in vivo investigations have reported that the cannabinoid receptor (CB1–CB2) agonist WIN55212–2 provided neuroprotection in animal models of neonatal HIE [324]. However, the production of psychoactive effects limits their therapeutic value. The phytocannabinoid, cannabidiol (CBD), is the major non-psychoactive component of *Cannabis sativa* plant (marijuana) [325]. CBD is well-known as a potent anti-inflammatory and antioxidant substance which shows neuroprotection in several neurodegenerative disorders using experimental animal models [326,327,328].

The neuroprotective effect of CBD against neonatal HIE was first demonstrated by Alvarez et al. [329]. It was reported that administration of CBD to newborn piglets at a dose of 0.1 mg/kg after HI insult leads to a reduction of brain edema and prevention of brain seizures. Also, CBD reduced cell loss and the number of degenerated neurons and showed additional cardio-protective effects. These effects can be attributed to a CBD-induced modulation of cerebral hemodynamic impairment and improvement of brain metabolic activity. Also, CBD did not show any significant side effects in treated piglets. Castillo et al. examined the effect of CBD (100 µM) in an in vitro model of neonatal HIE by exposing forebrain slices from newborn mice to oxygen-glucose deprivation [330]. The prevention of necrotic (reduction of LDH efflux to the medium) and apoptotic (reduction of caspase-9 concentration in tissue) cell death by CBD was observed. CBD also modulates inflammation (reduced IL-6 and TNF-α production and COX-2 expression) as well as excitotoxicity modulation (decreased glutamate release). Furthermore, iNOS expression was decreased. The results demonstrated that CB2 and adenosine—mainly A2A-receptors—participated in CBD-induced neuroprotection and led to activation of A2A, which was a principal mechanism of CBD neuroprotection. However, these reports demonstrated short-term neuroprotective activities of CBD in the immature brain after HI. In one study, Pazos et al. demonstrated long-term neuroprotective effects of CBD in newborn rats that received 1 mg/kg CBD after HI [331]. In this study, CBD reduced brain infarct volume and modulated brain excitotoxicity, inflammation, and oxidative stress 7 days following HI. It was reported that CBD provided long-lasting neuroprotection and treated animals’ neurobehavioral performance improved one month after HI. Also, they did not observe any noticeable side effects with CBD administration. In one study, Lafuente studied long-term (72 h after HI) effects of CBD. They found that post-HI i.v. injection of 0.1 mg/kg CBD in newborn piglets protects neurons, reduces cell death, and preserves astrocytes, which is related to its anti-inflammatory effect. Also, CBD improved neurobehavioral performance, brain activity, and stabilized metabolic activity [331]. In a fallow-up study by Pazos et al. [332], they investigated the underlying mechanisms in in vivo CBD-induced neuroprotection using the 1–2-day-old piglets treated with 1 mg/kg 30 min after HI. Findings indicated that CBD modulated excitotoxicity, inflammation, and oxidative stress, and both CB2 and 5HT1A receptors were involved in CBD-induced neuroprotection. In a recent study [333], the authors combined CBD with hypothermia in newborn piglets (receiving 30 min after the insult) and observed that the protective effect of combined therapy on excitotoxicity (glutamate/Nacetyl-aspartate ratio), inflammation (TNFα), oxidative stress (oxidized protein levels), and on cell damage was greater than either hypothermia or CBD alone, which indicated that CBD and hypothermia act complementarily if used shortly following the insult. Similarly, Barata et al. [38] observed the additive neuroprotective effects for combining hypothermia and CBD in newborn pigs (received 1 mg/kg) in a longer follow-up study that resulted in more complete neuroprotection than CBD or hypothermia alone. According to the results, CBD can be applied as a promising and novel therapy for neonates with HIE.

#### 4.2.18. Huperzine A

Huperzine A (HupA), a naturally occurring alkaloid which was firstly separated from the Chinese herb *Huperzia serrata*, is a potent AChE inhibitor [334]. It has been widely demonstrated that this agent represented learning- and memory-enhancing efficacy in animal and human studies. It also reversed or attenuated cognitive deficits in Alzheimer’s disease and other forms of dementia in animal experiments [335,336]. Besides, HupA also exerts multiple neuroprotective activities, for example it demonstrated neuroprotective properties, against hydrogen peroxide-, β-amyloid-, and glutamate-induced neurotoxicity and oxygen–glucose deprivation [337,338,339]. Furthermore, HupA has also been revealed to have neuroprotective activities against cerebral ischemic injury [340].

The protective activity of huperzine A against HIBD in neonatal rats was investigated by Wang et al. [341]. In this study, HI resulted in working memory impairments in pups after 5 weeks. Pups received huperzine A (0.1 mg/kg i.p.) once a day for 5 weeks after HI. They observed significant behavioral and histopathological protection against HI injury in rats treated with HupA. Furthermore, administration of HupA also decreased neuronal damage in ipsilateral hemisphere and provided remarkable protection in the hippocampal CA1 region. HupA showed its pharmacological activities by enhancing synaptic ACh levels via inhibition of AChE.

## 5. Conclusions

Despite the wide range of studies concerning neuroprotective drugs, currently, the only clinically relevant strategy to treat neonates with HI brain injury is hypothermia, but it is not able to reduce severe disabilities and is not completely successful, as approximately half of the affected neonates do not respond to it. Furthermore, hypothermia is expensive, and its therapeutic window is narrow. Thus, development of alternative and more effective therapeutic strategies is urgently needed.

The evidence presented in this review demonstrates the potential of natural plant products, especially polyphenols, in both prevention (agents administered before HI insult or through maternal consumption) and treatment (those administered after HI event) of neonatal HI brain damage, however regulated clinical trials have not been performed.

These compounds target both the early phase as well as later phase of injury via the following mechanisms: antioxidative, such as GSE, GSPE, and vanillin; anti-apoptosis, such as quercetin, oxymatrine, and caffeine; ameliorating blood barrier dysfunction, such as vanillin and vitexin; preventing microgliosis and inflammatory response, such as pterostilbene, etc. Moreover, some of these agents also regulate cell survival or cell death signaling pathways (Figure 2). For example, geniposide activates the PI3K/Akt cell survival signaling pathway or the neuroprotective effect of vitexin is related to early inhibition of HIF-1α. Many of these agents act not only on direct targets, but also on signaling pathways (see Table 1 and Figure 2). Since HIBD involves a series of complex progressive pathophysiologic processes, therefore, drugs acting on multiple targets or combinations of single target agents may be more effective in treating perinatal HIE.

When a HI event occurs, a substantial damage has taken place immediately after HI in the brain. Therefore, treatments have to start as early as possible in order to reduce further injury and promote regeneration. However, the preventive strategies will likely be the most effective therapeutic approaches. In this context, those compounds or extracts that act through maternal dietary supplementation, such as pomegranate juice and PIC, would be very interesting because nutritional approaches can easily be adopted as a preventive strategy in humans.

In conclusion, development of neuroprotective drugs from natural plant products is a prosperous approach for prevention and treatment of HI brain injury in neonates. Results of the presented studies may be useful for this purpose. In the future, more attention should be paid to further investigate the neuroprotective mechanisms, bioavailability, and metabolism of these agents, especially in humans.

## Figures and Tables

**Figure 1 ijms-22-00833-f001:**
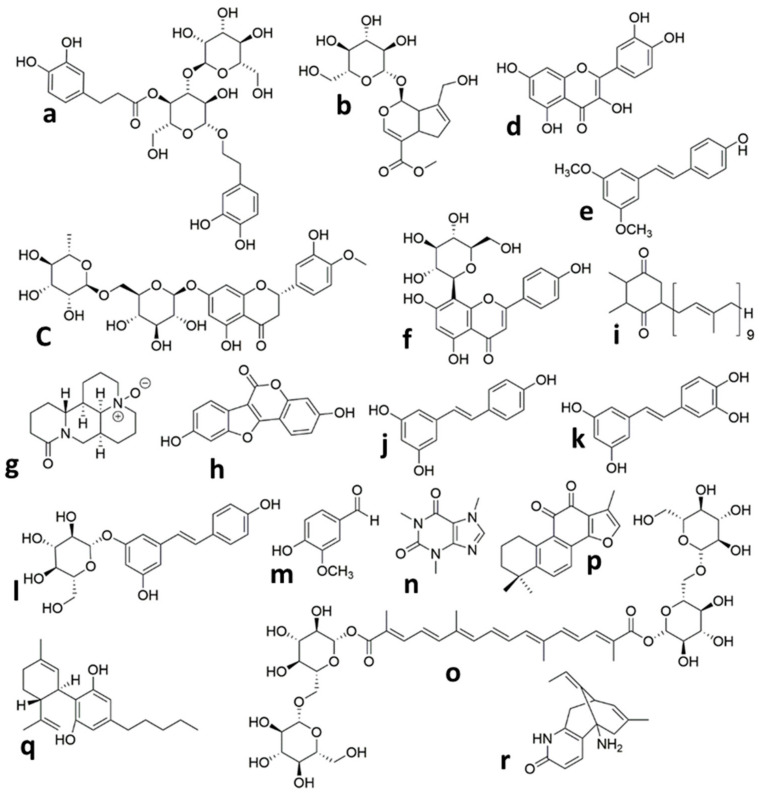
Chemical structures of the isolated compounds from plants in animal models of neonatal Hypoxic Ischemic brain injury. (**a**) Verbascoside, (**b**) Geniposide, (**c**) Hesperidin, (**d**) Quercetin, (**e**) Pterostilbene, (**f**) Vitexin, (**g**) Oxymatrine, (**h**) Coumestrol, (**i**) Plastoquinone, (**j**) Resveratrol, (**k**) Piceatannol, (**l**) Polydatin, (**m**) Vanillin, (**n**) Caffeine, (**o**) Crocin, (**p**) Tanshinone IIA, (**q**) Cannabidiol, (**r**) Huperzine A.

**Figure 2 ijms-22-00833-f002:**
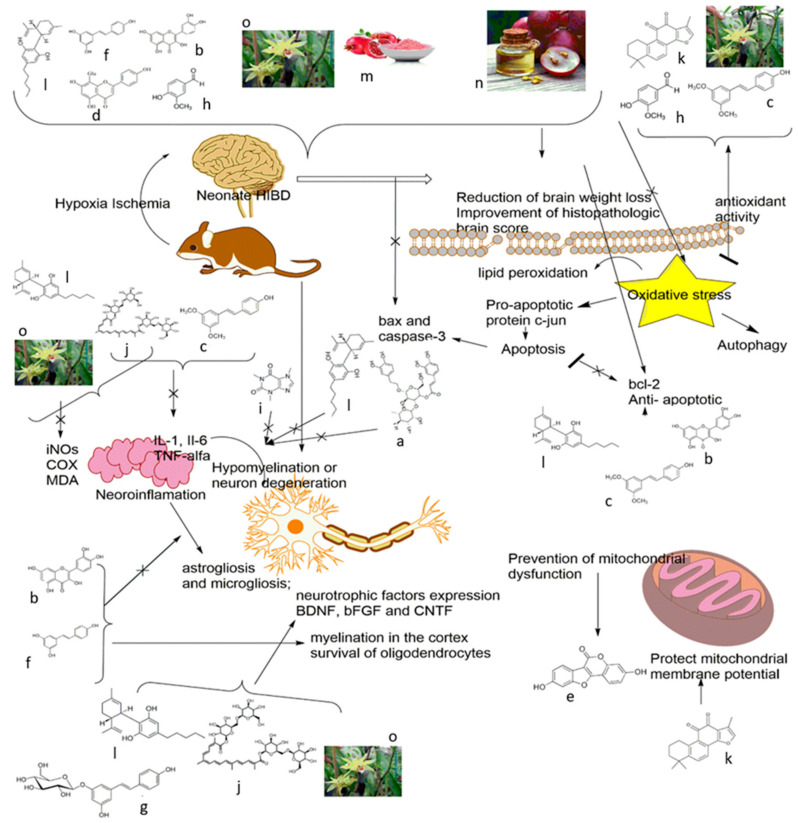
The excitatory, free radical toxicity, inflammation, and neuronal cell death mechanism of plant extracts/plant-derived compounds in animal models of neonatal hypoxic-ischemic brain injury. (**a**) Verbascoside, (**b**) Quercetin, (**c**) Pterostilbene, (**d**) Vitexin, (**e**) Coumestrol, (**f**) Resveratrol, (**g**) Polydatin, (**h**) Vanillin, (**i**) Caffeine, (**j**) Crocin, (**k**) Tanshinone IIA, (**l**) Cannabidiol, (**m**) Pomegranate juice and Pomegranate polyphenol extract, (**n**) Grape seed extract and Grape seed proanthocyanidin extract, (**o**) *Dendribium officinale.*

**Table 1 ijms-22-00833-t001:** The studied plant extracts/plant-derived compounds in animal models of neonatal hypoxic-ischemic brain injury.

Plant Extract/Compound	Administration Regimen	Dose	Species	Synergistic/Complementary Mechanism with/without Hypothermia	Effects/Molecular Mechanism	Reference
Grape seed extract	Pretreatment 5 min before HI Post-treatment 4 h after reoxygenation	50 mg/kg intra peritoneal (i.p.) injection twice daily for 1 day	P7 neonatal rat pups		Pretreatment: Reduction of brain weight loss Improvement of histopathologic brain score Antioxidant effect: Reduction of lipid peroxidation (reduced 8-iso-PGF2α and TBARS level) Post-treatment: No protection	[52]
	Post-treatment 5 min to 5 h after reoxygenation	50 mg/kg i.p. injection	P7 neonatal rat pups		Reduction of brain weight loss Antioxidant effect: Reduction of lipid peroxidation (reduced 8-iso-PGF_2α_ level) Anti-apoptotic effect: Reduction of pro-apoptotic protein c-jun	[53]
Grape seed proanthocyanidin extract	Pretreatment 20 min before HI	30 mg/kg i.p. injection	P7neonatal mouse pups		Reduction of infarct volume Improvement of neurobehavioral recovery Anti-apoptotic effect: Reduction of apoptotic cells number Reduction of pro-apoptotic proteins expression(bax and cleaved-caspase-3) Enhancement of anti-apoptotic (bcl-2) proteins expression and increased ratio of bcl2-to-bax	[60]
Pomegranate juice	Maternal dietary supplementation	1:60 or 1:80 or 1:320 dilutions of pj concentrate, during last third of pregnancy and litter suckling	P7 neonatal mouse pups		Reduction of brain tissue loss Anti-apoptotic effect: Reduction of caspase-3 activation	[36]
Pomegranate polyphenol extract	Maternal dietary supplementation	4.8 mg/day during pregnancy and litter suckling	P7 neonatal mouse pups		Anti-apoptotic effect Reduction of caspase-3 activation	[96]
*Dendribium officinale* extract	Post-treatment	(75 or 150 or 300) mg/kg, intragastric administration, Daily for 14 consecutive days	P7 neonatal rat pups		Improvement of behavioral ability Reduction of CI area percentage Regulation of brain injury-related proteins’ expression (diminished HDAC1 and elevated KCC2 expression) Antioxidant effect: Diminished levels of NOS, NO, and MDA Enhancement of antioxidant capacity (increased SOD activity) Anti-apoptotic effect Downregulated caspase-3 and Bax expressionUpregulated Bcl-2 expression Anti-inflammatory effect: Regulation of neurotrophic factors expression (increased expression of BDNF, bFGF, and CNTF) Mechanism: Inhibition of HIF-1α	[37]
Caffeine	Pretreatment immediately before HI, post-treatment 0, 24, 48, and 72 h after HI	20 mg/kg	P7 neonatal rat pups		Decreased ventriculomegaly and hypomyelination More normally arranged myelinated axon orientation Enhanced proportion of immature oligodendrocytes Anti-apoptotic effectDecreased number of apoptotic cells	[303]
Pterostilbene	Pretreatment 30 min before HI	50 mg/kg i.p. injection	P7 neonatal rat pups		Enhancement of animal survival Reduction of neurological score Improvement in motor coordination, motor deficit, and working memory deficit Reduction of brain infarct volume Reduction of brain edema Anti-inflammation: (decreased TNFα, IL-1, and IL-6 expression; decreased p-65 and NF-κβ expression) Antioxidant: (decreased TBARS and ROS level; increased SOD activity) Antiapoptotic: (inhibition of programmed cell death) Mechanism: HO-1 signaling pathway (Prevention of HO-1 mRNA and protein expression)	[172]
Quercetin	Pretreatment	40 mg/kg/day Intra-gastrically injection once a day for 7 days, last administration was 2 h after HI	P7 neonatal rat pups		Anti-apoptosis: (downregulated Bax, upregulated Bcl-2, attenuated DNA-strands breaks) Anti-inflammation: (attenuated astrogliosis and microgliosis; downregulated IL-6, IL-1β, and TNF-α) Mechanism: Suppression of TLR4-mediated NF-κB signaling pathway (decreased TLR4 phosphorylation and downstream signals of p65 and p-IκBα)	[159]
	Post-treatment 2 h after HI to the day of rats’ scarification	20 or 40 mg/kg Intra-gastrically injection Once a day	P3 neonatal rat pups		Improvement of myelination in the cortex through strengthening survival of oligodendrocytes	[160]
Resveratrol	Pretreatment 24 h/10 min before HI Post-treatment3 h after HI	20 mg/kg or 200 μg/kg or 2 μg/kg i.p. injection	P7 Neonatal rat and mouse pups		Pretreatment: Protecting brain from tissue loss Preventing apoptosis and necrosis (diminished caspase-3 and calpain activation) Post-treatment: No protection	[96]
	Pretreatment 30 min before HI Post-treatment 30 min after HI	30 mg/kg i.p. injection	P7 neonatal rat pups		Pretreatment: anti-apoptotic: (reduced expression level of Bax, caspase-3, and the ratio of Bax/Bcl-2) Post-treatment Not protection	[273]
	Post-treatment Immediately after HI	90 mg/kg i.p. injection	P7 neonatal rat pups		Reduced infarct volume Preserved myelination Behavioral tests (P8–P66): Improved early reflexes and sensorimotor functions Improved learning/memory function	[274]
	Pretreatment 10 min before HI Post-treatment Immediately after HI	20 mg/kg i.p. injection	P7 neonatal rat pups		Pretreatment: Reduced infarct volume Preserved myelination Minimized astroglial reactive response Improved long-term cognitive impairment Post-treatment: Not protection	[275]
	Post-treatment 0 h, 8 h and 18 h after HI	100 mg/kg i.p. injection	P7 neonatal rat pups		Anti- inflammation:(reduction of inflammatory factors expression; inhibition of microglia activation) anti-apoptosis: (regulation of Bax, Bcl-2 and caspase3 expression)	[276]
	Post-treatment immediately after HI and 12 h later	100 mg/mL i.p. injection	P7 neonatal mouse pups		Anti- inflammation: (reduced expression levels of IL-1β, IL-6, and TNF-α) Enhancement of SIRT1 levelMechanism: Inhibition of TLR4/MyD88/NF-κB signaling pathway (reduced TLR4, MyD88, NF-κB levels)	[277]
Tanshinone IIA	From 2 days before HI for 9 or 16 days	10 mg/kg/day i.p. injection	P7 neonatal rat pups		Improved neuropathology and sensorimotor functions Improved antioxidant capacity Protect mitochondrial membrane potential	[320]
Vanillin	Post- treatment Immediately after HI	80 mg/kg, i.p. injection	P7 neonatal rat pups		Improved neurobehavioral deficits Reduced brain infract volume Reduced brain edema Reduced neuronal degeneration and necrotic cell death Preserving BBB integrity (decreased IgG leakage, reduced levels of MMPs, upregulated expression of TJ-related proteins) Inhibiting oxidative stress (increased SOD, GSH-Px, and CAT; increased T-AOC; decreased lipid peroxidation (MDA content))	[297]
Verbascoside	Post-treatment	(60 or 120 or 240) mg/kg, i.p. injection, every 12 h for two consecutive days	P7 neonatal rat pups		Reduction of prolonged reflex latencies Reduction of brain infarct volume Reduction of necrosis Reduction of degeneration and morphological damage of neurons Reduction of autophagosome formation Reverse expression level of autophagy-related proteins (Beclin-1, LC3-II/I ratio, and P62)	[115]
Vitexin	Post-treatment 5 min or 3 h after HI	(30 or 45 or 60) mg/kg i.p. injection single dose	P7 neonatal rat pups		5 min after HI:Reduction of brain infarct volume Reduction of brain edema Improvement of long-term behavioral development Attenuation of neuronal cell death and brain tissue loss Attenuation of BBB disruption: (reduction of IgG staining and brain water content) Mechanism: inhibition of HIF-1α/VEGF(decreased HIF-1α and VEGF protein levels) 3 h after HI: vitexin lost its neuroprotection	[188]
Cannabidiol	Post-treatment 15 and 240 min after HI	0.1 mg/kg i.v. injection	P3–P5 newborn piglets		Reduction of brain edema Prevention of brain seizures Reduced cell lossReduced number of degenerating neurons Cardio-protective effects	[329]
	Post-treatment	100 µM	Forebrain slices from P7–P10 neonatal mouse pups		Prevention of necrotic cell death (reduced LDH efflux to the medium) Anti-apoptotic: (reduced caspase-9 level) Anti-inflammatory: (reduced IL-6 and TNF-α level; reduced COX-2 expression level) Excitotoxicity modulation: (decreased glutamate release) Antioxidant: (decreased iNOS expression) Mechanism: Activation of A2A and CBD receptors	[330]
	Post-treatment 15 and 240 min after HI	0.1 mg/kg i.v. injection	P1–P3 newborn piglets		Long follow-up study (72 h after HI): Improved neurobehavioral performance Improved brain activity Stabilized metabolic activity Reduced cell death Anti-inflammatory effect (protects astrocytes)	[331]
	Post-treatment 10 min after HI	1 mg/kg i.v. injection	P7–P10 neonatal rat pups		Long follow-up study (7 days or one month after HI): Reduced infarct volume Modulated brain excitotoxicity, oxidative stress, and inflammation 7 days after HI Improved neurobehavioral performance one month after HI	[332]
	Post-treatment 30 min after HI	1 mg/kg i.v. injection	P1–P2 newborn piglets		Modulation of excitotoxicity, oxidative stress and inflammationMechanism Involvement of CB2 and 5HT1A receptors	[333]
	Post-treatment 30 min after HT Combined with hypothermia	1 mg/kg i.v. injection	P1–P2 newborn piglets	Short follow-up study: Modulation of excitotoxicity, inflammation, oxidative stress, and cell damage was greater than either hypothermia or CBD alone	Cannabidiol treatments reduced the oxidized protein and TNFα levels	[38]
	Post-treatment 30 min after HT Combined with hypothermia	1 mg/kg i.v. injection repeated 24 and 48 h after HI	P1 newborn piglets	Reverse the induction of increases in lactate/N-acetyl-aspartate, glutamate/Nacetyl-aspartate, TNFα, or apoptosis Modulate the brain excitotoxicity and inflammation More complete neuroprotection than CBD or hypothermia alone	Long follow-up study: Improved brain activity Decreased microglial activation	[334]
Coumestrol	Pretreatment Immediately pre-hypoxia Post-treatment 3 h after hypoxia	20 mg/kg i.p. injection	P7 male neonatal rat pups		Pre- and post-treatment: Preventing reference and working memory impairments Preventing late cognitive deficits in reference and working memory at P60 Reduction of brain tissue loss Reduction of late GFAP overexpression (index of late reactive astrogliosis) Pretreatment: Prevention of mitochondrial swelling Mechanism: Prevention of mitochondrial dysfunction and long-term cognitive deficits	[224]
Crocin	Post-treatment Immediately or 2 h after HI	10 mg/kg combined with hypothermia	P7 neonatal mouse pups	Reduced neurological severity score Anti-oxidative effect: (reduced levels of MDA, ROS, and NO; reduced mRNA expression of iNOS and COX-2) Anti-inflammatory effect: (reduced IL-1β and TNF-α levels)		[311]

## Data Availability

Not applicable.

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
