# Peer review of "A Review of Plant Extracts and Plant-Derived Natural Compounds in the Prevention/Treatment of Neonatal Hypoxic-Ischemic Brain Injury"

_ijms, 2021, doi:10.3390/ijms22020833_

Round 1

Reviewer 1 Report

  • This is a comprehensive review of the potential natural neuroprotective agents that could be trialled in neonatal hypoxic-ischemic injury models.

  • Needs some grammar edits and typos corrected
  • Be consistent with tense (of language used in text)
  • Would rephrase text referring to this review as a study. But highlight comprehensive review of ...
  • Between main headings, further subheadings would be helpful to break up the text and help guide the reader (Section 2 and 3).

More general to be checked throughout:

  • Check and be consistent with references; ie author (surname) et al., (year) found... (no author first name).
  • Possibly useful to briefly mentioned the animal models used (that the bulk of the studies reviewed use). The focus is mostly mouse/rat HI models, with a mention of piglets
  • Be careful with use of HI Haemorrhage, do you mean ischemic injury? Different terms, please check throughout
  • As the table is well detailed, it could be helpful to re-order some bits to highlight when treatments where given. Pre-treatments would be working on different mechanisms of action versus post-treatment etc. also highlighting which have been trialled with hypothermia.
  • Potentially important to highlight synergistic / complementary mechanisms these would then be working in (with/without hypothermia)
  • Would change reference to mouse/rat neonates to just pups (or neonatal rat/mouse pups)
  • The intro and conclusion allude to preterms and term but the bulk of the review focuses on term (P7 rat) so it would be better to just focus on term HI to avoid and confusion

More specifically,

  • P5; was it brain weight loss or reduced volumetric injury?
  • P6 line 272: ‘science observed..’ please rephrase
  • Section 3 title: potential mechanisms for neuroprotective strategies?
  • The table is extensive, maybe a figure would also be of use? Time course, potential points of intervention?

Author Response

Reviewer 1

This is a comprehensive review of the potential natural neuroprotective agents that could be trialled in neonatal hypoxic-ischemic injury models.

  • Needs some grammar edits and typos corrected
  • The grammar errors and typographical errors were corrected in all part of the manuscript.
  • Be consistent with tense (of language used in text)
  • All tense in the sentences were checked and corrected.
  • Would rephrase text referring to this review as a study. But highlight comprehensive review of ...
  • The abstract line 20 and 21 changed to “there is a critical need for reviewing the effective therapeutic strategies to explore the protective agents and methods.”
  • Between main headings, further subheadings would be helpful to break up the text and help guide the reader (Section 2 and 3).
  • The section 2 was sub headed into 4 part “Excitatory, Free radical toxicity, Inflammation and Neuronal cell death”. The section 3 was sub headed into 3 part. Part 1 is “Reduction of cellular apoptosis and oxidative stress in HI”, part 2 is “Reduction of neurotrophic-evoked excitation and inflammatory cytokines in HI” and part 3 is “Animal model of HI”.

More general to be checked throughout:

  • Check and be consistent with references; ie author (surname) et al., (year) found... (no author first name).
  • All the Reference were checked and rearranged again
  • Possibly useful to briefly mentioned the animal models used (that the bulk of the studies reviewed use). The focus is mostly mouse/rat HI models, with a mention of piglets
  • The section 3. Animal model of HI was added to the main text.
  • Be careful with use of HI Hemorrhage, do you mean ischemic injury? Different terms, please check throughout
  • HI Hemorrhage in all part change to HI insult.
  • As the table is well detailed, it could be helpful to re-order some bits to highlight when treatments where given. Pre-treatments would be working on different mechanisms of action versus post-treatment etc. also highlighting which have been trialled with hypothermia.
  • It was shown in Table as the Administration regimen, the Effects/molecular mechanism parts and also every kind of treatment was explained in front of its own administration regimen with its mechanism. Additionally the column synergistic / complementary mechanisms related to hypothermia was added to table
  • Potentially important to highlight synergistic / complementary mechanisms these would then be working in (with/without hypothermia)
  • It was explained in the Administration regimen part that which phytochemicals has the synergistic effect with hypothermia and additionally the column synergistic / complementary mechanisms related to hypothermia was added to table
  • Would change reference to mouse/rat neonates to just pups (or neonatal rat/mouse pups)
  • Neonatal rat/mouse pups based on the study was added to the species part of the Table.
  • The intro and conclusion allude to preterms and term but the bulk of the review focuses on term (P7 rat) so it would be better to just focus on term HI to avoid and confusion
  • The introduction and conclusion part has been change in introduction part line 34 and 35 “perinatal” and “both preterm and full-term” and in discussion part line 1615 “Unfortunately, the use of hypothermia is possible only in full-term neonates, however modern neonatology is able to care for severely premature babies.” have been deleted to show the better concept of the using natural product on term animals.

More specifically,

  • P5; was it brain weight loss or reduced volumetric injury?
  • According to the Yangzheng Feng and et all study, it was reduced brain weight loss.

  • P6 line 272: ‘science observed..’ please rephrase.
  • It was changed to “The neuroprotective effects were observed”.
  • Section 3 title: potential mechanisms for neuroprotective strategies?
  • The word “mechanisms for” was added in section 3 title
  • The table is extensive, maybe a figure would also be of use? Time course, potential points of intervention?
  • The figure with caption “The Excitatory, Free radical toxicity, Inflammation and Neuronal cell death mechanism of plant extracts/plant-derived compounds in animal models of neonatal hypoxic-ischemic brain injury” was added and the “synergistic / complementary mechanism with/without hypothermia” column was added to the Table.

Reviewer 2 Report

The article “Natural products for the prevention/treatment of neonatal hypoxic-ischemic brain injury: A pharmacological review”, provides a comprehensive review of the literature around the mechanisms and natural intervention approaches that have emerged as candidate therapies for HIE in recent years. This review catalogs an extensive array of natural products and their application to neonatal hypoxic-ischemic brain injury. This appears to be first review paper to assemble such a wide-ranging set of natural candidate therapeutics for HIE.

Major Comments:

  • Although the article is generally clear with the significance of the topic well defined, there are numerous grammatical and word choice issues throughout that at times limit readability and clarity. This is particularly true when describing details of specific studies. Lines 264-279 need carful rereading and significant grammatical updating due to incomplete sentence structure and incorrect grammar usage. Issues noted in this paragraph can be seen throughout the text and should be updated to ensure clear communication of the cited sources and their findings.

  • On line 259, the authors should use caution when citing animal models of “depression” as this has become an increasingly controversial topic. Specifically, animal models of psychiatric conditions, such as depression, are most commonly induced through stress exposure. A distinction needs to be made between observable behaviors in animal models such as, anhedonia, motor slowing, social withdrawal versus the specific clinical classification of “depression”, which includes states that are not observable or measurable in animal models. At the very least, the authors should use the term “depression related” models or remove reference to depression altogether as it has limited reliance to the HIE focus of the manuscript.

Minor Comments:

  • Line 267, authors should replace the use of “semester” with a more appropriate term related to the study design by Loren et al., 2005. Consider including the actual window of time that pomegranate juice was administered.

Author Response

Reviewer 2

The article “Natural products for the prevention/treatment of neonatal hypoxic-ischemic brain injury: A pharmacological review”, provides a comprehensive review of the literature around the mechanisms and natural intervention approaches that have emerged as candidate therapies for HIE in recent years. This review catalogs an extensive array of natural products and their application to neonatal hypoxic-ischemic brain injury. This appears to be first review paper to assemble such a wide-ranging set of natural candidate therapeutics for HIE.

Major Comments:

  • Although the article is generally clear with the significance of the topic well defined, there are numerous grammatical and word choice issues throughout that at times limit readability and clarity. This is particularly true when describing details of specific studies. Lines 264-279 need carful rereading and significant grammatical updating due to incomplete sentence structure and incorrect grammar usage. Issues noted in this paragraph can be seen throughout the text and should be updated to ensure clear communication of the cited sources and their findings.
  • It was checked and grammatical errors and incomplete sentence were corrected in all part of the manuscript

  • On line 259, the authors should use caution when citing animal models of “depression” as this has become an increasingly controversial topic. Specifically, animal models of psychiatric conditions, such as depression, are most commonly induced through stress exposure. A distinction needs to be made between observable behaviors in animal models such as, anhedonia, motor slowing, and social withdrawal versus the specific clinical classification of “depression”, which includes states that are not observable or measurable in animal models. At the very least, the authors should use the term “depression related” models or remove reference to depression altogether as it has limited reliance to the HIE focus of the manuscript.
  • The term “depression related” models was used.

Minor Comments:

  • Line 267, authors should replace the use of “semester” with a more appropriate term related to the study design by Loren et al., 2005. Consider including the actual window of time that pomegranate juice was administered.
  • It was explained precisely that “The treatment time for groups was in most cases 15 day that includes 7 dayin utero and 8 day ex utero for mice that sustained caspase-3 analysis and 21 day that includes 7 day in utero and 14 day ex utero for mice that sustained histologic analysis.” 

Round 2

Reviewer 2 Report

The article “Natural products for the prevention/treatment of neonatal hypoxic-ischemic brain injury: A pharmacological review”, provides a comprehensive review of the literature around the mechanisms and natural intervention approaches that have emerged as candidate therapies for HIE in recent years. 

I thank the authors for making the revisions that have satisfied my original review requests.